# Measuring Sodium from Discretionary Salt: Comparison of Methods

**DOI:** 10.3390/nu15245076

**Published:** 2023-12-12

**Authors:** Rachael Mira McLean, Nan Xin Wang, Claire Cameron, Sheila Skeaff

**Affiliations:** 1Department of Preventive & Social Medicine, University of Otago, Dunedin 9016, New Zealand; 2Department of Human Nutrition, University of Otago, Dunedin 9016, New Zealand; nanxin.wang@otago.ac.nz (N.X.W.); sheila.skeaff@otago.ac.nz (S.S.); 3Biostatistics Centre, Division of Health Sciences, University of Otago, Dunedin 9016, New Zealand; claire.cameron@otago.ac.nz

**Keywords:** sodium, salt, discretionary salt, New Zealand, 24 h urine, lithium-tagged salt, dietary assessment

## Abstract

(1) Background: The best method to assess discretionary salt intake in population surveys has not been established. (2) Methods: This secondary analysis compared three different methods of measuring sodium intake from discretionary salt in a convenience sample of 109 adults in New Zealand. Participants replaced their household salt with lithium-tagged salt provided by researchers over eight days. Baseline 24 h urine was collected, and two further 24 h urine and 24 h dietary recalls were collected between days six and eight. Discretionary salt was estimated from the lithium-tagged salt, focused questions in the 24 h dietary recall, and the ‘subtraction method’ (a combination of 24 h urine and 24 h dietary recall measures). (3) Results: Around one-third of estimates from the ‘subtraction method’ were negative and therefore unrealistic. The mean difference between 24 h dietary recall and lithium-tagged salt estimates for sodium from discretionary salt mean were 457 mg sodium/day and 65 mg/day for mean and median, respectively. (4) Conclusions: It is possible to obtain a reasonable estimate of discretionary salt intake from careful questioning regarding salt used in cooking, in recipes, and at the table during a 24 h recall process to inform population salt reduction strategies.

## 1. Introduction

The World Health Organization (WHO) has identified reducing population salt intake as one of nine important strategies to reduce noncommunicable diseases (NCDs), including elevated blood pressure. It set a target of reducing population salt intake by 30% by 2025 in its Global Action Plan for reducing NCDs in 2013 [1]. Progress has been slow [2], despite a package of evidence-based interventions outlined in the SHAKE technical package [3]. The WHO recommends that a range of strategies be implemented to reduce adult salt intake from two main sources: salt in processed food and discretionary salt (salt that is added by the consumer in cooking and at the table).

Increasing dietary potassium is also important to control blood pressure, aiming for a sodium:potassium molar ratio of less than one [4,5]. Potassium-enriched reduced-sodium salt replacement products (also referred to as low-sodium salt substitutes) have been identified as a useful tool to decrease sodium and increase potassium intakes [6]. A Cochrane systematic review and meta-analysis of 20 randomized controlled trials showed that the use of reduced-sodium salts (most of which were potassium enriched) was associated with an average reduction in blood pressure of 4.76 mmHg systolic and −2.43 mmHg diastolic in adults [7]. Most of the evidence regarding the use of reduced-sodium salts relates to their use as a discretionary salt replacement, with a few studies examining their use in processed food such as bread [7,8].

Discretionary salt is that which is added by the consumer or household during cooking and at the table. Understanding the sources of salt intake is essential to inform strategies relating to dietary sodium and potassium. The proportion of total salt intake from discretionary sources varies considerably by country and by dietary pattern. For example, in parts of China, discretionary salt makes up to around 70% of total salt intake, whereas in a more typical Western diet, it is generally around 15% of total intake, with the majority of intake from salt already added to processed food during manufacturing [9].

Understanding the amount of discretionary salt and the proportion of total salt intake from discretionary sources is important for informing a range of strategies relating to dietary sodium, potassium, and iodine. Where discretionary salt is a large proportion of total intake, sodium reduction strategies include a strong focus on educating consumers to reduce the amount of salt they use and on replacing salt with reduced-sodium salts containing potassium. On the other hand, where the majority of salt intake is from processed food, reformulation is a key strategy. Salt reduction and replacement strategies must also align with iodine strategies, especially where discretionary salt is fortified with iodine and in places where there is a history of iodine deficiency [10].

Measuring discretionary salt as a proportion of total intake at a population level has methodological challenges. The lithium-tagged salt method is considered the gold-standard method of measuring discretionary salt and involves participants using lithium-tagged salt over a study period (usually around a week), during which time multiple 24 h urine samples are collected [11,12,13]. Studies using the lithium-tagged salt method tend to be small due to the high respondent burden involved [11,12]. Twenty-four-hour urine collection is the gold-standard method of assessing total salt (sodium) intake, as around 90% of ingested salt is excreted in the urine over a 2 -h period [14]. However, 24 h urine samples are burdensome to collect and analyze for both participants and researchers [15], and they do not yield information about sources of salt intake without the use of lithium-tagged salt. Twenty-four-hour dietary recalls are often used in population dietary surveys. They have been shown to be less accurate as a measure of total salt intake, but they can provide valuable information regarding food sources of salt [16,17]. However, dietary recalls seldom include questions designed to quantify discretionary salt consumption [15].

A variety of methods were used to estimate discretionary salt as a proportion of total dietary salt intake in a recent systematic review of population-based studies [9]. Of 33 studies identified in the systematic review that quantified this proportion, only 2 used the lithium-tagged salt method, while 15 used a 24 h dietary recall and 9 used a food diary. Five studies based their estimates on sales data, which are unlikely to be accurate at an individual level. Rhodes et al. [18] showed that the USDA automated multiple-pass method for 24 h dietary recall is a valid measure of total sodium intake at the population or group level compared with 24 h urinary excretion; however, they did not quantify discretionary salt added at the table. Similarly, the National Health and Nutrition Examination Surveys (NHANES) 2003–2012 used dietary habit questions to assess discretionary salt intake (for example, ‘How often do you add ordinary salt to food at the table? Would you say rarely, occasionally, very often, refused, don’t know?’) to assess discretionary salt intake [19]. They were therefore able to report habits relating to discretionary salt use but not the amount of discretionary salt consumed. As more countries introduce population dietary salt reduction strategies globally [2], it is important that accurate estimates of discretionary salt as a proportion of total salt intake be produced.

We undertook a study that measured the amount and proportion of discretionary salt intake in New Zealand using several methods, including the use of lithium-tagged salt. Discretionary salt had not previously been measured in New Zealand, a country where salt intake is high, yet iodine intake has been low over many years [20]. This paper is a secondary analysis of data from this study. Here, we aim to compare the lithium-tagged salt method with two other measures: specific questions as part of a 24 h dietary recall and the subtraction method (outlined below) [21].

## 2. Materials and Methods

This paper is a secondary analysis. A detailed description of the methods has been published elsewhere [22]. Briefly, the study recruited 116 healthy volunteers between 18 and 40 years old in a single city in New Zealand. We produced lithium-tagged salt in our laboratory using standard methods [11] to a known concentration. Participants were instructed to collect a baseline 24 h urine sample (Day 1). Starting the following day, participants replaced their usual salt intake with lithium-tagged salt for seven days. In total, participants were given 55g of lithium-tagged salt in a small container and a saltshaker. Between day 6 and day 8, participants collected another two 24 h urine samples. A 24 h dietary recall was conducted a day after each 24 h urine collection (excluding baseline). In total, three 24 h urine samples and two 24 h dietary recalls were collected for each participant.

### 2.1. 24 h Urine Collection

Participants were given a pack that included collection bottles and a jug. They were instructed to discard the first morning’s urine on the day of collection and note the time. They then collected all the urine they passed into the provided bottle until the first morning urination of the following day, again noting the time. They were also asked to record any times they missed a collection during the 24 h period. Incomplete 24 h urine samples were determined using three criteria: self-report of two or more missing void samples in 24 h, total urine volume less than 500 mL, or 24 h urinary creatinine excretion falling outside the reference range (4.0 to 17.0 mmol/day for females and 7.0 to 24.0 mmol/day for males) [23]. At baseline, three participants had 24 h urine samples that were considered incomplete. Overall, 109 participants with complete baseline urine were included in the study, as 4 participants did not complete the study protocol. [22]. Urinary sodium was analyzed using the ion-selective electrode method [24] on a Roche Cobas C311 system.

### 2.2. 24 h Dietary Recall

Face-to-face multiple-pass 24 h dietary recalls based on methods developed by the United States Department of Agriculture (USDA) were conducted [18,25]. In addition, participants were asked specifically to report their salt use in cooking or at the table. For each cooked food item, participants were asked if they had added salt during cooking. If they had added salt, they were asked to specify the quantity of salt added in ‘shakes’ or teaspoons. For each meal, participants were asked if they had added salt at the table and to specify the quantity of salt added. If the participant used the saltshaker, they were asked how many shakes of salt they added. For example, when a participant reported that they consumed home-cooked pasta, they were asked if salt was added during cooking or at the table. Salt added from the provided pottle was estimated with household measures such as measuring spoons. Each shake of salt was standardized and recorded as 0.06 g of salt. Three researchers independently weighed each shake of salt ten times, and the average weight of each shake of salt was used as the standardized weight. The 24 h dietary recalls were then entered into Xyris FoodWorks 10 (Xyris Pty Ltd., Queensland, Australia) using the New Zealand FOODfiles 2018 [26]. The amount of sodium intake from discretionary salt was summed from the amount of discretionary salt that participants reported consuming in each collected 24 h dietary recall.

### 2.3. Lithium-Tagged Salt Method

Lithium excretion in 24 h urine was used to determine sodium intake from discretionary salt (mg/day). Lithium excretion at baseline was used to correct for lithium intake that did not come from the provided lithium-tagged salt [27]. The following steps were used to calculate sodium intake from discretionary salt:

Step 1: Lithium excretion (day 6 or 8)—lithium excretion at baseline = corrected lithium excretion (mg/day);

Step 2: Corrected lithium excretion (mg/day) ÷ concentration of lithium measured in lithium-tagged salt (i.e., 1.42 mg/kg or 1.36 mg/kg) = discretionary salt intake (g/day);

Step 3: Discretionary salt intake (g/day) ÷ 400 (1 g of sodium chloride contains 400 mg of sodium) = sodium intake from discretionary salt (mg/day).

### 2.4. Indirect “Subtraction” Method

The indirect “subtraction” method was used to determine discretionary salt by calculating total sodium intake assessed using 24 h urine sample minus intake from processed foods (nondiscretionary salt) from the 24 h recall [28].

### 2.5. Determining Mean Sodium Intake from Discretionary Salt for Each Method

Sodium intake from discretionary salt (mg/day) for all methods was reported as the mean of the two 24 h urine collections if both urine samples collected between days 6 and 8 were regarded as complete collections. However, there were 13 participants who had only one complete urine collected between days 6 and 8. For these participants, data from the one complete sample were used.

### 2.6. Statistical Analysis

Participants’ demographics were summarized descriptively using means and standard deviations for continuous variables and numbers and percentages for categorical variables. We assessed sodium from discretionary salt intake in three ways:The lithium-tagged salt method, which assumes that any lithium measured in the 24 h urine after 5 days of use is from discretionary salt used over the same 24 h period. This was our reference method because it is widely considered to be the gold standard;From the dietary recall questions on discretionary salt use;The subtraction method [21], which is calculated as 24 h urine total sodium minus sodium intake from processed foods from the 24 h recall.

Pearson correlation (r) and scatter plots were used to assess the relationship and the agreement between the gold-standard lithium-tagged salt method and the other two methods (24 h dietary recall and indirect “subtraction” method). A Bland–Altman plot [29] was produced in order to measure agreement between results from the lithium-tagged salt method and 24 h dietary recall measures. All statistical analyses were conducted using Stata version 17.0 [30].

## 3. Results

Baseline characteristics of participants were recorded (Table 1); around 50% were female. Total sodium and energy intake appeared to be similar to results reported for population dietary surveys in New Zealand [31,32], even though we did not seek to have a sample that was representative of the population overall.

Table 2 shows results of sodium intake from discretionary salt in mg/day and as a proportion of total salt intake from the three methods. Mean sodium intake from discretionary salt using the lithium-tagged salt method was 537 mg/day, with a 95% confidence interval (CI) of 415, 658 mg/day (Table 2).

Figure 1 shows a scatter plot demonstrating the relationship between sodium from discretionary salt estimated using the subtraction method and the reference method, i.e., lithium-tagged salt. Around one-third of estimates made using the subtraction method are below 0 (negative values). Pearson’s correlation coefficient for this relationship is 0.36. While negative values of discretionary salt are unrealistic, the subtraction methods allows negative values to be calculated.

Figure 2 shows a scatter plot demonstrating the relationship between sodium intake from discretionary salt estimated using the 24 h dietary recall and the reference method, i.e., lithium-tagged salt. Pearson’s correlation coefficient for this relationship is 0.32.

Figure 3 demonstrates the degree of agreement between the sodium intake from discretionary salt estimated by 24 h dietary recall and the reference method using a Bland–Altman plot [29]. This plot shows that at lower levels of intake, the results are in relative agreement, although this is not the case at higher levels. The mean difference calculated using this method is −457 mg/day (95%CI −709, −205). This plot demonstrates that there are a few outliers that have a large influence on the mean difference and that both measures are highly skewed.

## 4. Discussion

We found that the use of focused questions in a 24 h dietary recall is a more accurate measure of discretionary salt intake than the ‘subtraction method’ compared to the gold-standard lithium-tagged salt method in this sample of 109 New Zealand adults. Although the subtraction method uses the best measures of both total salt intake (24 h urine) and salt intake from purchased food (24 h dietary recall), one-third of values calculated using the subtraction method were negative, which is unrealistic. Therefore, we tested both the correlation and agreement between the results obtained from focused questions in the 24 h recall and the reference method, showing that these questions enabled a reasonable estimate of discretionary salt intake, particularly at lower levels of intake.

The questions that were asked regarding discretionary salt intake in our study were devised by the research team and go beyond what is usually asked in a dietary survey. The standard USDA multiple-pass tool does not quantify salt added at the table and quantifies salt content in home-cooked dishes using standardized recipes rather than individually [18]. The face-to-face 24 h recall collection enabled us to adapt the standard method and probe participants on their use of discretionary salt. However, questions relating to discretionary salt use could be added to an automated collection tool. In interpreting the results of these questions, several assumptions were made. We relied on a standard estimate of what a ‘shake of the saltshaker’ is based on testing of the saltshakers provided in the study by the research team. This may be highly variable in different contexts and countries where different methods of adding salt at the table and in cooking are likely [2]. We also assumed that when vegetables or other foods such as pasta were cooked in water with added salt, the water was discarded prior to serving. While this is common practice in New Zealand, it may not be the case in other countries, where vegetable and cooking water may be included in subsequent cooking or sauces or salt is not widely added. Therefore, determination of discretionary salt in 24 h dietary recall is likely to differ depending on the country and the context.

Although the 24 h dietary recall method was a better measure than the subtraction method, the mean estimate in grams per day for 24 h dietary recall is almost double that of the lithium-tagged salt method, with a mean difference 457 mg sodium/day. However, the median estimates are much closer, with a mean difference of only 65 mg/day, indicating that the measures are highly skewed. Our results are similar to those reported in a study undertaken in rural Guatemala [13], where nine pairs of mothers and their sons aged 6–9 years replaced their household salt with lithium-tagged salt for nine days and collected four 24 h urine samples, in addition to duplicate portions of food over three days and three 24 h recalls. The study compared the results of discretionary salt estimates from the lithium-tagged salt method with those from duplicate portions and 24 h dietary recall, showing that 24 h recalls overestimated discretionary salt intake by 90% for the mothers and 185% for the sons, with even greater overestimates for the duplicate portion method [13]. Both the amount of discretionary salt and the proportion of total salt intake that was discretionary salt were substantially higher among the Guatemalan study participants than in our study. This may have increased the observed differences, as our study showed that the differences between estimates were greater with higher discretionary salt intake (Figure 2 and Figure 3).

Our finding of greater differences at higher levels of discretionary salt intake contrasts with findings from validation studies relating to other foods. A validation study by Lafrenière et al. in which self-reported portion sizes using an automated web-based 24 h recall in the French language were compared to measured portions in a feeding study showed a greater error rate among portions smaller than 100 g than among those above 100 g [33]. The mean estimation errors reported by Lafrenière et al. were 17.1% for portions under 100 g compared to only −2.4% for portions over 100 g, also showing that smaller portions were overestimated in terms of size, whereas larger portions were underestimated. The contrast between the findings of Lafrenière et al. and those reported in our study may be due to the fact that salt is added as a condiment rather than a food, and it is consumed in much smaller quantities (all below 6 g/day in our study).

Reasonable estimation of discretionary salt as a proportion of total salt intake is essential for designing public health interventions to reduce sodium intake and increase potassium intake. In particular, the likely impact of replacing ordinary table salt with potassium-enriched salt substitutes is determined by the amount of discretionary salt used. Several studies have shown that the use of potassium-enriched salt substitutes in place of ordinary table salt (sodium chloride) can be effective and is associated with important reductions in blood pressure and cardiovascular disease [7]. The Salt Substitute and Stroke Study was a cluster randomized controlled trial conducted in rural China; the intervention involved replacing table salt with a salt substitute comprising 25% potassium chloride and 75% sodium chloride. In this high-risk population, after a mean followup period of 4.74 years, use of the salt substitute was associated with reductions in dietary sodium intake, systolic blood pressure, and incidence of stroke, as well as increased potassium intake [34]. A cluster randomized controlled trial conducted in elderly care facilities in China demonstrated that when potassium-enriched salt substitutes were used instead of table salt in cooking and at the table, blood pressure was lowered, and there were fewer cardiovascular events in the intervention group. Twenty-four-hour urinary excretion of potassium was increased, but there was no change in 24 h sodium excretion [35]. Similarly, a cluster randomized controlled trial conducted in Peru showed that use of a salt substitute was associated with reduced blood pressure and increased potassium intake; however, no evidence of reduced sodium intake was reported [8]. The use of salt substitutes is likely to have been particularly effective in these studies, as both China and Peru have a large proportion of total salt intake from discretionary salt [9,36].

The use of salt substitutes has several potential advantages. Once household salt has been replaced with a salt substitute, the intervention does not require further behavior change with respect to food preparation and salt use at the table. This strategy does not require engagement with the food industry to reformulate processed food or modify food labelling, which can be difficult and time-consuming for governments and other agencies [6].

However, use of potassium-enriched salt substitutes is likely to be of limited effectiveness in communities where the majority of salt intake is from processed foods and discretionary salt is only a small proportion of total salt intake [6]. Therefore, estimates of both the amount and proportion of discretionary salt (as a proportion of total salt intake) are essential to inform population public health strategies in this area. Using the gold-standard lithium-tagged salt method in a small non-representative sample may not reflect population-level behavior in this regard. Our study shows that focused questions tailored to the local population can be included in 24 h diet recall, producing reasonable estimates.

Given the recent evidence regarding the potential effectiveness of potassium-enriched salt substitutes, it is vital that governments understand the proportion of salt that comes from discretionary salt in their populations. Twenty-four-hour dietary recall is commonly used to assess intakes of a variety of nutrients in both population and epidemiological studies [9]. Our research shows that including questions about discretionary salt as part of a 24 h recall can provide valuable information about the amount and proportion of salt that comes from discretionary sources. This can inform population salt reduction strategies. However, the specific questions used need to be trialed in different populations depending on local cooking methods. For example, it is important to know how salt is added during cooking (by teaspoon or handful) and whether cooking water is discarded or added to other dishes.

The degree to which a precise estimate of population discretionary salt intake is necessary depend on the purpose of collecting this information. If the aim is to inform population salt reduction programs, then a highly accurate measure is less important than one that is more easily achievable in a representative sample of the population. Furthermore, when considering whether to highlight discretionary salt as an important source of salt in the diet, it is more important to know if the proportion of discretionary salt is 10%, 40%, or 80% than whether the proportion is 10% or 12%. If the aim is to determine how discretionary salt contributes to iodine intake (to inform iodine fortification strategies), other ways to monitor intake, such as spot urine iodine collection, may need to be undertaken in target populations.

This study has several strengths, including the use of the gold-standard lithium-tagged salt method as a reference; the high study retention rate; and the adherence to study protocols. Limitations include the potential for over- or under-collection of 24 h urine collection, which always exists in such studies, even when a biomarker of completeness is used [15]. These results may not be generalizable to other populations due to the variety of patterns of use of discretionary salt.

## 5. Conclusions

It is possible to obtain a reasonable estimate of discretionary salt intake from careful questioning regarding salt used in cooking, in recipes, and at the table during a 24 h recall process to inform population salt reduction strategies. Although the lithium-tagged salt method is considered a ‘gold standard’, it is burdensome and unlikely to be undertaken in large representative population samples.

## Figures and Tables

**Figure 1 nutrients-15-05076-f001:**
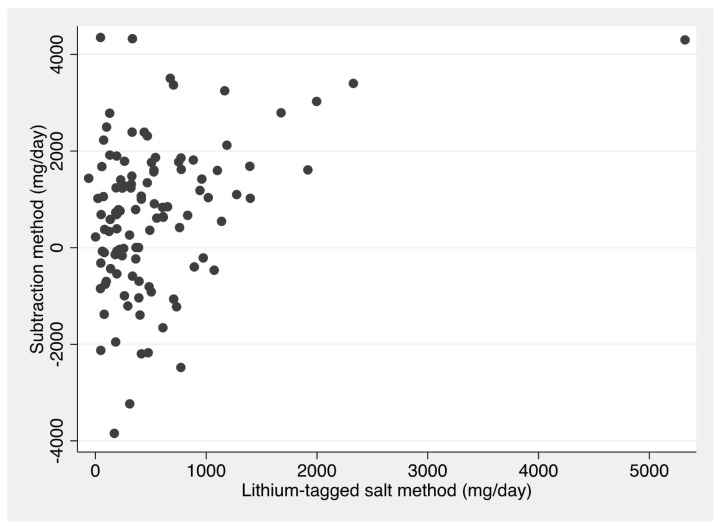
Scatter plot of sodium from discretionary salt (mg/day) estimated using the lithium-tagged salt method (reference) vs. the subtraction method.

**Figure 2 nutrients-15-05076-f002:**
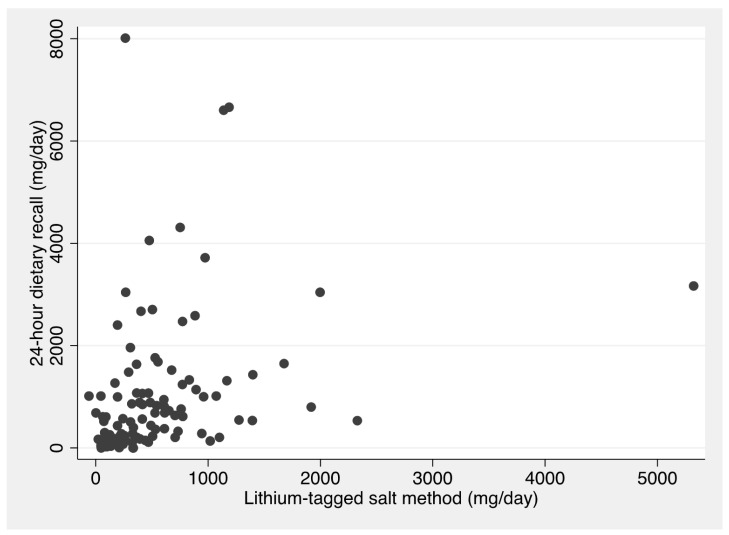
Scatter plot of sodium from discretionary salt (mg/day) estimated the lithium-tagged salt method (reference) vs. 24 h dietary recall.

**Figure 3 nutrients-15-05076-f003:**
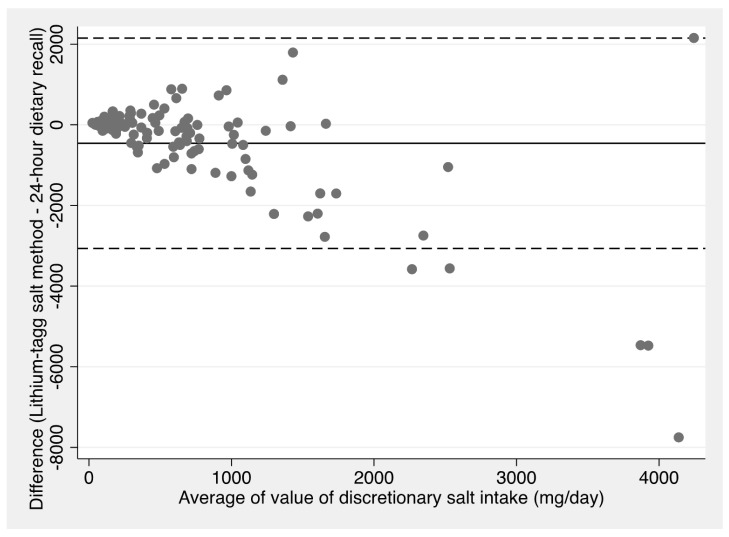
Bland–Altman plot of sodium from discretionary salt estimated using the lithium-tagged salt method vs. 24 h dietary recall. Mean difference: −457 mg/day (95%CI −710, −205).

**Table 1 nutrients-15-05076-t001:** Baseline Characteristics of Participants.

Characteristic	Total (n = 109)
Age (years), mean (SD)	26.0 (6.1)
Sex, n (%)	
Female	57 (52)
Male	52 (48)
BMI (kg/m^2^), mean (SD)	24.3 (4.7)
Energy intake (kJ/day) (24 h dietary recall), mean (SD)	10,142 (3027)
Mean followup sodium intake (mg/day) (24 h dietary recall), mean (SD)	3838 (1844)
Mean followup 24 h urinary sodium excretion (mg/day), mean (SD)	3300 (1233)

**Table 2 nutrients-15-05076-t002:** Estimated sodium intake from discretionary salt from the three methods.

Method	Sodium from Discretionary Salt (mg/day)		
	Mean (SD)	Median (25th, 75th Percentile)	95% Confidence Interval of the Mean
Lithium-tagged salt method	537 (642)	366 (186, 705)	415, 658
24 h dietary recall	994 (1385)	546 (173, 1074)	731, 1257
Subtraction method	675 (1545)	761 (−233, 1611)	382, 968
Difference			
Lithium-tagged salt method—24 h dietary recall	−457 (1330)	−65 (−546, 86)	
Lithium-tagged salt method—subtraction method	−138 (1443)	−221 (−1026, 597)	

## Data Availability

The raw data supporting the conclusions of this article will be made available by the authors upon reasonable request.

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
