# Peer review of "Measuring Sodium from Discretionary Salt: Comparison of Methods"

_nutrients, 2023, doi:10.3390/nu15245076_

Round 1

Reviewer 1 Report

Comments and Suggestions for Authors

The manuscript makes a potentially important contribution to the research literature, although the authors should address the matters below.

The use of a convenience sample seems reasonably under the circumstances. However, with such a sampling design it is important to establish that the usual assumptions attesting to the validity of the findings are satisfied--normality, homoscedasticity, linearity.

The findings seem important for purposes of policy and practice. Some practical implications are drawn from the findings, but the impact of the manuscript can be broadened considerably by providing more detail about such implications.

A very small point is that "Person" is used in place of "Pearson" on line 157.

Comments on the Quality of English Language

Only routine editing seems in order. The document does not seem to have any serious readability issues.

Author Response

Reviewer 1.

The manuscript makes a potentially important contribution to the research literature, Thank you 

although the authors should address the matters below.

The use of a convenience sample seems reasonably under the circumstances. However, with such a sampling design it is important to establish that the usual assumptions attesting to the validity of the findings are satisfied--normality, homoscedasticity, linearity. 

It is reassuring that reviewers have these statistical considerations at front of mind.  However, this is not a convenience sample, per se, this is a group of people who were recruited to take part in our CATS trial – so they met very particular criteria in order to participate.  Nevertheless, it is important to consider the assumptions of the methods used.  We have calculated means and confidence intervals, used Pearson’s correlation coefficient and produced Bland Altman plots.  Means and confidence intervals assume the sampling distribution of the mean is normally distributed.  The Central Limit Theorem tells us that 109 participants provides a large enough sample to assume normality.  Pearson’s correlation coefficient is assessing whether there is a linear relationship between two variables.  Bland Altman is a graphical display of the relationship between the difference in estimates and the average of the estimates – again looking at the relationship (not necessarily linear) between characteristics of the data.  The normality assumption is not relevant to these last two tests. Linearity is what we are assessing with correlation, so we are not assuming linearity.  Heteroscedacity is something that, again, we would assess through these tests so is not an assumption.  Unequal variance would be illustrated in the plots. Thanks for allowing us to clarify these points.  We have not added any of this to the manuscript because these are standard tests.

The findings seem important for purposes of policy and practice. Some practical implications are drawn from the findings, but the impact of the manuscript can be broadened considerably by providing more detail about such implications. 

We have added further details about the implications for policy and practice (lines 310-319)

A very small point is that "Person" is used in place of "Pearson" on line 157. 

Thank you I have changed this.

Reviewer 2 Report

Comments and Suggestions for Authors

Thanks for giving me the opportunity to review this paper. The authors compare various methods for the comparison of discretionary sodium intake. 

The paper is well presented and sound, but the authors should provide the questionnaire L210 they used for transparency. If needed it should be provided as supplementary material.

Table 1: the mean energy intake is 10 142 (please provide the unit). I guess these are not kCal. Please provide unites everywhere.

L207 the authors explain that it is possible to have reasonnable estimates, espacially at lower levels of intake. This apparently contradicts Lafrenière, J., Lamarche, B., Laramée, C. et al. Validation of a newly automated web-based 24-hour dietary recall using fully controlled feeding studies. BMC Nutr 3, 34 (2017). doi: 10.1186/s40795-017-0153-3, who stated in this paper 'However, in contrary to our initial assumptions, we observed that portions smaller than 100 g were estimated at a greater error rate than those of 100 g and above'. 

The authors should clarify their finding and put it in perspective of the work of Lafreniere et al 2017.

Other than that nice job and best of luck with the publication process.

Author Response

Reviewer 2

Thanks for giving me the opportunity to review this paper. The authors compare various methods for the comparison of discretionary sodium intake. 

The paper is well presented and sound, but the authors should provide the questionnaire L210 they used for transparency. If needed it should be provided as supplementary material. 

We have added further details about the specific questions included in the face to face dietary recall (lines 131-135), and 230-234.

Table 1: the mean energy intake is 10 142 (please provide the unit). I guess these are not kCal. Please provide unites everywhere. 

Thank you We have added units to Table 1.

L207 the authors explain that it is possible to have reasonable estimates, especially at lower levels of intake. This apparently contradicts Lafrenière, J., Lamarche, B., Laramée, C. et al. Validation of a newly automated web-based 24-hour dietary recall using fully controlled feeding studies. BMC Nutr 3, 34 (2017). doi: 10.1186/s40795-017-0153-3, who stated in this paper 'However, in contrary to our initial assumptions, we observed that portions smaller than 100 g were estimated at a greater error rate than those of 100 g and above'.  

The authors should clarify their finding and put it in perspective of the work of Lafreniere et al 2017.

Please see paragraph  lines 262-271

Other than that nice job and best of luck with the publication process.

Thank you.